# Chronic Pulmonary Aspergillosis Situation among Post Tuberculosis Patients in Vietnam: An Observational Study

**DOI:** 10.3390/jof7070532

**Published:** 2021-06-30

**Authors:** Ngoc Thi Bich Nguyen, Huy Le Ngoc, Nhung Viet Nguyen, Luong Van Dinh, Hung Van Nguyen, Huyen Thi Nguyen, David W. Denning

**Affiliations:** 1Vietnam National Lung Hospital, Hanoi 10000, Vietnam; vietnhung@yahoo.com (N.V.N.); dinhvanluong66@gmail.com (L.V.D.); hungmtb75@gmail.com (H.V.N.); huyennguyen0406@gmail.com (H.T.N.); 2Manchester Academic Health Science Centre, Faculty of Biology, Medicine and Health, University of Manchester, Manchester M23 9LT, UK; 3Global Action Fund for Fungal Infections, 1208 Geneva, Switzerland

**Keywords:** chronic pulmonary Aspergillosis, prior tuberculosis, Vietnam, developing countries

## Abstract

This study provides a brief view of chronic pulmonary aspergillosis (CPA) in the post-tuberculosis treatment community in Vietnam, a high burden tuberculosis (TB) country. In three months in late 2019, 70 post-TB patients managed at Vietnam National Lung Hospital were enrolled. Of these, 38 (54.3%) had CPA. The male/female ratio was 3/1 (28 males and ten females). CPA patients had a mean age of 59 ± 2.3 years (95%CI 54.4–63.6). The mean Body mass index (BMI) was 19.0 ± 0.5 (18.0–20.0) and 16 of 38 (42.1%) patients had concurrent diseases, the most common of which were chronic obstructive pulmonary disease (COPD) and diabetes. Twenty-six patients (68.4%) developed hemoptysis, 21 (55.3%) breathlessness, and weight loss was seen in 30 (78.9%). Anaemia was seen in 15 (39.5%) and 27 of 38 (71.1%) patients had an elevated C-reactive protein (CRP). The most common radiological findings were multiple cavities (52.6%) and pleural thickening (42.7%), followed by aspergilloma (29.0%) and non-specific infiltrates. There were five of 38 patients (13.2%) with a cavity containing a fungal ball on the chest X-ray, but when the high resolution computed tomography (HRCT) was examined, the number of patients with fungal balls rose to 11 (28.9%). Overall, 34 of 38 (89.5%) cases had an elevated *Aspergillus* IgG with an optical density ≥ 1, and in 2 cases, it was 0.9–1.0 (5%), borderline positive. In nine patients (23.7%) *Aspergillus fumigatus* was cultured from sputum. CPA is an under-recognised problem in Vietnam and other high burden TB countries, requiring a different diagnostic approach and treatment and careful management. HRCT and *Aspergillus* IgG serum test are recommended as initial diagnostic tools for CPA diagnosis.

## 1. Introduction

TB remains a worldwide health problem with complicated sequelae even after the end of treatment [1]. Among TB sequelae, chronic pulmonary aspergillosis (CPA) is the most deadly chronic sequela with multiple presentations [2]. CPA can supervene in many respiratory diseases, including chronic obstructive pulmonary disease (COPD), asthma, and cystic fibrosis, affecting an estimated 3 million people. TB and probably COPD patients are the most common predisposing conditions. About one among five post-TB patients with a cavity will develop CPA after treatment [3].

This disease silently destroys lung parenchyma with progressive cavitation, pleural thickening, and fibrosis. The usual lack of specific symptoms and a definitive diagnostic test leads to high mortality rates, even higher with antifungal drug resistance [4,5].

In limited resources settings and high burden TB countries, CPA diagnosis is commonly confused with other diseases, such as smear-negative TB. These errors lead to harmful problems in both clinical and epidemiological settings: the slow process of diagnosis and treatment, low treatment adherence, and poor outcomes increases the public health risk of developing multidrug-resistant TB because of lack of confidence in TB services, and waste of healthcare resources [5]. Hence, a good approach and understanding of the disease are essential, especially in high burden TB countries [6].

Vietnam is ranked 13th among 30 high burden TB countries [1]. Previous studies have reported a high burden of fungal infections and also antifungal drug resistance [7]. Despite these complications, direct CPA data are lacking in Southeast Asia and Vietnam. Hence, we conducted a study with two objectives: to describe the CPA situation among post TB patients and to evaluate the prognostic factors of having CPA after PTB, to give a preliminary view of CPA in Vietnam, a tropical country with a high TB burden and limited laboratory diagnostics for fungal disease.

## 2. Materials and Methods

### 2.1. Patients

A retrospective observational study was conducted with medical records of patients at Vietnam National Lung Hospital from October 2019 to December 2020. All the patients were diagnosed based on international guidelines from the European Society for Clinical Microbiology and Infectious Diseases and the European Respiratory Society [2,8].

All patients with a history of prior TB therapy who presented with abnormal radiological findings and prolonged respiratory symptoms were recruited for this study. The patients were diagnosed with CPA if they had the following criteria: (i) presenting with clinical features such as prolonged cough, weight loss, bronchiectasis, or hemoptysis lasting for at least three months; (ii) radiological findings suggesting any features of CPA, including aspergilloma, cavity, multiple cavities, pericavitary infiltrates, thickened pleura or fibrosis [9]; (iii) serological or microbiological evidence implicating *Aspergillus* spp. or histopathological evidence [10].

We excluded patients with either a history of having antifungal treatment in the last two months before the clinical review or active TB.

### 2.2. Clinical Samples and Radiological Findings

Respiratory specimens, including bronchoalveolar lavages (BAL) and sputum, were cultured and followed a standard protocol provided by our Vietnam National Reference TB Laboratory, located in our hospital, to ensure quality handling. The description of our National Reference Laboratory center can be found elsewhere [11].

Positive culture for *Aspergillus* sp. from a respiratory sample (BAL, sputum) and histopathology confirmed were taken as solid evidence of CPA. All patients were also tested for *Mycobacteria* spp. using GeneXpert or culture in sputum and BAL to exclude any TB relapse.

All the patients were screened with a chest X-ray and also had a high solution computed tomography (CT) scan of the thorax. An expert panel interpreted the chest X-rays and CT scans with experienced pulmonologists and radiologists for consensus on the characteristic features of CPA or other conditions.

### 2.3. Serology Test

We tested serum IgG *Aspergillus* antibodies in all the patients using the *Aspergillus fumigatus* IgG ELISA (enzyme immunoassay (EIA)) (Bordier Affinity Products, Switzerland). The specimens’ investigation was done as per the manufacture’s recommendations with an optical density (OD) cut-off of 1 for a positive IgG serum level. Moreover, we analyzed with the cut-off OD value of 0.9 to find differences between the IgG serum levels [12].

### 2.4. Treatment

CPA treatment followed international guidelines (400 mg oral itraconazole (2 capsules) daily after a meal for at least six months in those who agreed to take it. We have not collected follow-up data on response (a future study).

### 2.5. Statistical Analysis

We used the STATA^®^ version 13 (STATA, College Station, TX, USA) for data management and analysis. The demographic and other features were analyzed by descriptive analysis with mean, median, and percentages. All the tests with a *p*-value < 0.05 were considered significant.

### 2.6. Ethical Issues

This study was approved by the Vietnam National Lung Hospital Ethical Committee and all the recruited patients signed a consent form before study enrollment.

## 3. Results

In total, 70 patients were referred to our hospital because of respiratory symptoms, having been treated for TB previously. Among them, 40 (56.7%) patients were diagnosed with CPA. Two CPA patients with active TB were excluded. Finally, we had 38 CPA patients with a history of prior TB. The male/female ratio was 3/1 (28 males and 10 females). CPA patients had a mean age of 59 ± 2.29 years (95%CI 54.35–63.64) (minimum 22 years and maximum 86 years). Mean BMI was 18.98 ± 0.496 (17.97–19.98). For comorbidities, 16/38 (42.1%) patients had concurrent diseases. The most common were COPD and diabetes (Table 1). Several patients had bronchiectasis (three cases). Cultures for *Mycobacteria* were negative for all species, other than the two excluded above.

Ten patients had been treated for recurrent TB once, and three patients had been treated for TB three times. Remarkably, 14 of 33 (42.4%) patients had a history of being treated for TB more than 10 years previously and 12 (30.3%) 5–10 years previously. Two-third of CPA patients had previously treated TB more than seven years previously (the longest time is 31 years).

Productive cough was the primary symptom noticed in nearly 100% of patients in this study, often with mucus production (*n* = 37 (97.4%)). Twenty-six patients (68.42%) developed hemoptysis, 21 (55.3%) breathlessness, and weight loss was seen in 30 (78.9%). Anaemia was seen in about a third of patients—15 (39.5%).

Culture grew *Aspergillus fumigatus* in 10 cases, six in sputum, and four from BAL specimens. No other species were found. Histopathology findings were characteristic of CPA in three patients who underwent lung lobectomy and the other two who had transthoracic lung biopsies.

Overall, 27 of the 38 (71.1%) patients had an elevated C-reactive protein (CRP) (Table 2). All patients were tested for *Aspergillus* IgG (Table 2).

Overall, 34 of 38 (89.5%) cases had an *Aspergillus* IgG optical density ≥ 1, and in four cases, IgG was less than 1 (10.5%). The mean IgG OD was 2.82 ± 1.9. There were four cases with an *Aspergillus* IgG < 1. One case had lung lobectomy, so confirmed histopathologically. Others were confirmed with *Aspergillus* in sputum and BAL culture (in one case, the *Aspergillus* IgG OD was 0.978, and two others were 0.482 and 0.738). If the OD threshold was reduced to 0.9, two more cases were defined as CPA using serology (94.7%) (Figure 1).

We found a slight correlation between the duration of TB infection and *Aspergillus* IgG levels (*p*-value = 0.005) (Figure 2). However, the R squared is low (0.01), indicating that the height of the *Aspergillus* IgG antibody response is only slightly influenced by time.

With regard to the radiological findings, half of the patients had bilateral damage, 31.9% had damage on the right side, and the remainder only on the left. The most common findings were multiple cavities (52.6%) and pleural thickening (42.7%), followed by aspergilloma (29.0%) and infiltrates. Five of the 38 patients (13.2%) had a cavity containing a fungal ball on the chest X-ray, but when the HRCT was examined, the number of patients with fungal balls rose to 11 (28.9%). (Figure 3).

Following evaluation in clinic, and with all the results available to treating physicians, 21 (55.2%) patients were treated with itraconazole therapy. However, we did not collect follow-up information.

## 4. Discussion

CPA is a complicated pulmonary syndrome that requires a careful approach in diagnosis, treatment, and management. This is the first descriptive study from Vietnam to provide a brief view of the clinical spectrum and associated factors of CPA in the post-TB population. A recent attempt to quantify the likely prevalence of CPA in Vietnam estimated 55,500 affected people, 61 per 100,000, which is an internationally high rate. Post-tuberculous cavitation is common in Vietnam at 41% [13,14,15].

In this study, we highlighted the features/clinical symptoms of CPA among post-tuberculosis patients. This is the first study to describe the characteristics of CPA among the post-TB community in Southeast (SE) Asia. This study provides essential material for developing a national guideline for diagnosing, treating, and managing CPA.

Most of the patients we identified with CPA were older than 50, mean age 59 ± 2.25 years. This is similar to other studies, where CPA is most often reported mostly in middle-aged patients. In a previous study, age has been reported as a risk factor for poor outcomes in CPA TB patients [16]. The link with older age may be partially explained by the chronic development of CPA [3], but some comorbidities were also found; 10% of our patients had at least one comorbidity, mostly COPD or diabetes. Diabetes has been described as a significant association with CPA, as reported in several previous studies [16,17,18]. Diabetes is also considered a risk factor for developing a cavity during pulmonary TB and may be a risk factor for developing CPA [19,20,21]. In low-burden TB countries, other common risk factors are COPD and prior pulmonary surgery history [19,22]. However, in our study, only 20% of CPA patients had a COPD history. With a COPD prevalence of 6.7% and a population of 100 million [23], it is also essential to consider CPA as a differential diagnosis for TB among COPD patients in Vietnam. A holistic approach and screening for comorbidities in CPA patients is also required.

Furthermore, when screening people with a history of prior TB, we found two patients with positive TB tests who had concurrent CPA. Even with the exclusion of these patients due to the mimicking of symptoms and radiological findings, this interesting finding indicates that *Aspergillus* and *Mycobacterium* can combine to damage the lung. Hedayati in Iran and Iqbal in Pakistan reported that 13.7% and 13% of patients with CPA were co-infected with *M. tuberculosis,* respectively [18,24].

Hence, we recommend that every patient with PTB, active or with a prior history, should be screened for CPA to prevent the possibility of having a co-infection, especially if symptoms persist despite anti-tuberculous therapy.

Our data should raise awareness of CPA and the need to consider other respiratory diseases with similar non-specific symptoms. CPA is being underestimated and often confused with another chronic pulmonary disease, especially smear-negative TB [2]. The years of misdiagnosis of tuberculosis and incorrect treatment is addressable with clinical training and testing for fungal disease and aspergillosis.

In our study among post-TB patients, 56.7% were found to have only CPA and 2.9% had CPA and PTB, using criteria proposed for low resource settings [2]. This result is higher than similar studies in Uganda [6,25]. It can be explained by our small sample size. Globally, the prevalence of CPA among post TB patients ranged from 21–35%, depending on country situation [25]. What we cannot know is how many patients died of CPA, misdiagnosed as TB, and therefore could not have been referred for evaluation. The cohort seen in our center are probably the ‘slow progressors’, and much more aggressive CPA is likely to have been fatal [26].

Most of the CPA patients we identified developed CPA at least two years after treatment for PTB. The history of TB treatment is discussed in other studies [18]. Previous studies have reported higher percentages of having CPA after PTB treatment than patients just finishing complete therapy [3]. CPA development over time can explain this even after TB treatment. The interval between completing TB therapy and development of CPA is often many years. Suspicion and early diagnosis of CPA is important as five-year mortality rates are 50% even with treatment [2].

Prolonged cough with mucus and hemoptysis are the most common symptoms of CPA, followed by weight loss and fever [27]. Although cough and hemoptysis are typical, they are not reliable symptoms alone for CPA diagnosis as several other conditions including bronchiectasis and PTB have these manifestations. Symptoms lasting longer than three months is a clue for clinicians to think about CPA [2,12]. Otherwise, progression and death are common [28,29].

A CT scan is recommended as an initial diagnostic aid tool when CPA is clinically considered, providing better and more precise information than X-rays [30,31,32]. Over 50% of our patients had bilateral damage, and right-sided damage is more common than the left, consistent with the TB damage.

Unlike invasive pulmonary aspergillosis in which the specific halo sign can often be seen, the CPA radiological findings include aspergilloma, multiple cavities, and pleural thickening or necrotizing development [33]. Radiographic CT scan features secondary to *Aspergillus* infection range from a typical appearance of a fungus ball within the lung cavity to complex pleuro-parenchymal features related to the progressive destructive cavitary disease. Our study’s most common manifestation was multiple cavities, similar to other reviews [12,34]. Typical radiological findings have been described elsewhere [35,36]. It is difficult to distinguish CPA from other cavitary diseases such as tuberculosis, chronic cavitary pulmonary histoplasmosis, bronchiectasis, or even a combination of these conditions. In a few select cases, when *Aspergillus* develops into a fungal ball in a cavity, we can see an unusual type of CPA with a cavity with a crescent moon inside surrounding a fungal ball [35].

The role of *Aspergillus* IgG has been reported in previous studies [2,3,37]. This test is often considered the gold standard test for CPA diagnosis and is valuable for tracking the disease’s progression with a high positive predictive value [11,22]. In an Indonesian study, the *Aspergillus* IgG was a convenient tool with good diagnostic performance and simple application [3]. In most studies, *Aspergillus* IgG is positive in 80–92% of CPA cases, and we had two false negative tests in our patients.

An approach using clinical symptoms and *Aspergillus* IgG would be the initial screen for diagnosing CPA [38,39]. Our study also found a high level of IgG associated with a history of prior PTB, similar to previous studies, where active TB patients do not have an elevated IgG level [17]. As CPA can silently develop after completion of TB treatment, annual CPA evaluation including *Aspergillus* IgG to check seems appropriate.

Some of our patients were treated with antifungals: 400 mg itraconazole daily following international guidelines. This treatment is effective in some studies [40]. However, nearly half of the CPA patients did not receive any specific treatment. Our clinicians are poorly educated when approaching CPA patients, hence the poor uptake of self-funded antifungal therapy. More education and a national guideline is needed to raise the alertness of clinicians about fungal diseases.

Furthermore, in our country setting, a question arises about the azole-resistant situation since Justin Bearsley reported an extraordinary prevalence of drug-resistant *Aspergilli* in Vietnam [5]. Further studies should evaluate the frequency and risk of azole-resistance in the future.

Our study has some limitations. We only conducted a retrospective study, which had limited information and may suffer from recall bias. Another problem is the small number of patients, which may not be generalizable across the country. However, this study could be considered a cornerstone, along with previous studies, to raise awareness and improve clinical practice of CPA among the PTB community. Further research on the CPA situation in Vietnam is necessary to address potential problems such as drug-resistance in fungi.

## Figures and Tables

**Figure 1 jof-07-00532-f001:**
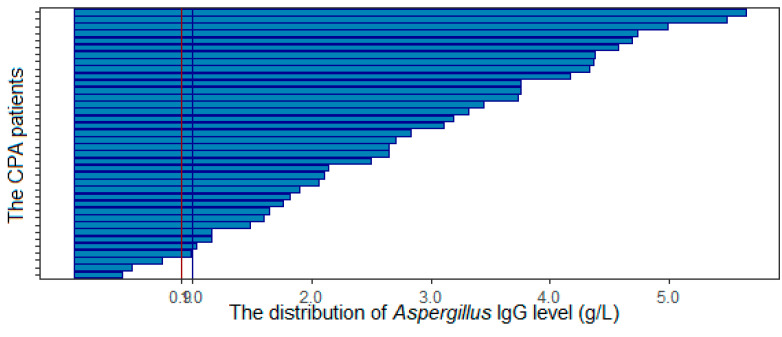
Summary of *Aspergillus* IgG distribution.

**Figure 2 jof-07-00532-f002:**
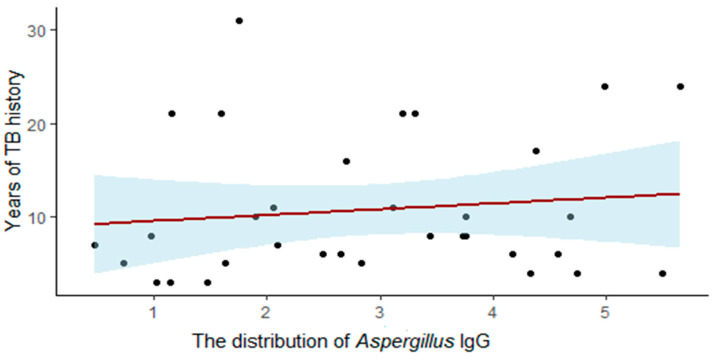
The correlation between *Aspergillus* IgG level and years of TB history.

**Figure 3 jof-07-00532-f003:**
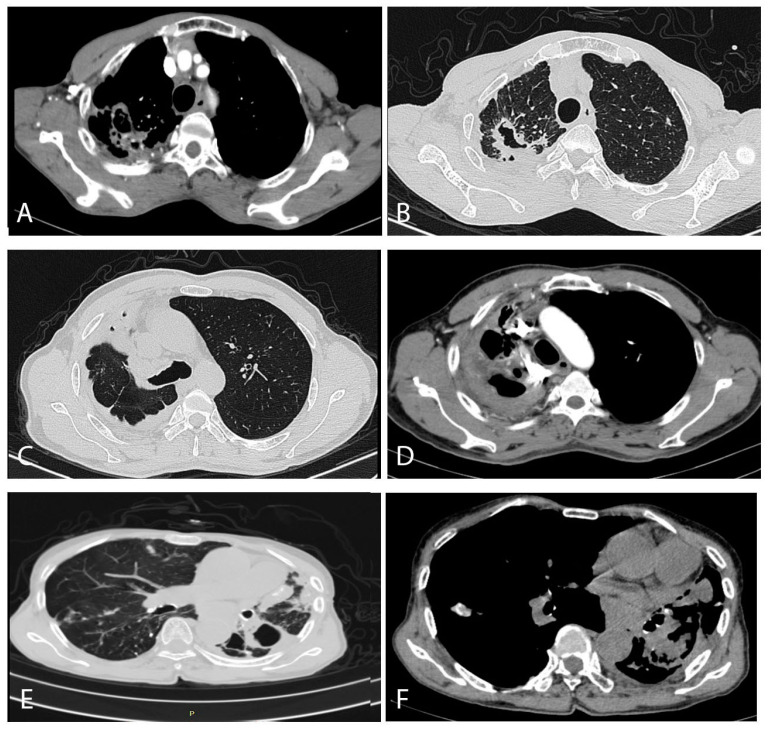
Typical computed tomography findings in our CPA patients. (**A**,**B**) were from a 71-year-old male patient, (**C**,**D**) were of a 50-year-old male patient, and (**E**,**F**) were from a 78-year-old female patient. (**A**) (contrast enhanced scan) shows enlarged arterial vessels on the edge of two separate cavities posteriorly in the right lung. The lung windows from a slightly higher section show a large thick-walled cavity with an irregular interior lining, and probably three other much smaller cavities, in association with remarkable pleural thickening posteriorly, with some pleural fat latero-posteriorly. No aspergilloma is visible in either image. (**C**) shows extensive pleuro-pulmonary fibrosis encasing the right upper lung, with two small cavities (probably) anteriorly. The right main bronchus and mediastinum is shifted to the right. In (**D**), slightly higher in the chest at the level of the aortic arch, shows considerable major arterial blood vessel distortion, additional enlarged arteries within the areas of inflammation or fibrosis and an anterior cavity. (**E**) shows at least one thick-walled cavity at the top of the left overlying an area of significant pleural thickening, with areas of consolidation or fibrosis anteriorly, containing some calcification on a bullous emphysematous background. There is a small area of ill-defined inflammation in the right lung. The bronchial walls contain significant calcification. (**F**) also shows the major mediastinal shift to the left, with extensive areas of consolidation or fibrosis with no particular pattern.

**Table 1 jof-07-00532-t001:** The demographic, clinical symptoms, and radiological findings of CPA patients.

Features	No.	Statistics
Baseline demographics		
CPA confirmed cases	40	
Age mean (SD)/Median (IQR)	59	2.3 (54.4–63.6)
Males	28	73.7%
Females	10	26.3%
BMI	19.0	0.49 (18.0–20.0)
Comorbidities		
Diabetes	5	12.5%
COPD	8	20%
Bronchiectasis	3	7.5%
Interval after TB to CPA presentation		
<5 years	9	27.3%
5–10 years	10	30.3%
>10 years	14	42.4%
Clinical symptoms		
Cough	37	97.4%
Productive cough	31	81.6%
Hemoptysis	18	47.4%
Dyspnea	21	55.3%
Fever	8	21.1%
Weight loss	14	36.8%
Chest X-ray radiological findings		
Cavitary lesion	8	21.1%
Aspergilloma	5	13.2%
Pleural thickening	30	79.0%
Hemithorax	5	21.1%
Chest CT findings		
Bilateral	19	50.0%
Left	7	18.4%
Right	12	31.6%
Multiple cavities with thickened pleura	20	52.6%
Fungal ball(s) (aspergilloma)	11	28.9%
Single cavity with thickened pleura	6	15.8%
Pleural thickening	17	44.7%
Bronchiectasis	10	26.3%
Non-specific infiltrates	11	28.9%

**Table 2 jof-07-00532-t002:** The laboratory parameters of CPA patients.

**Blood Test**	**Mean/Median**	**SD/IQR**	**Normal Range**
WBC	10.6	4.9	4.5–11. 0
CRP	60.9	71.7	0–5.0
Aspergillus IgG	2.82	1.9	<0.9

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
