# Peer review of "Chronic Pulmonary Aspergillosis Situation among Post Tuberculosis Patients in Vietnam: An Observational Study"

_jof, 2021, doi:10.3390/jof7070532_

Round 1

Reviewer 1 Report

This study looks at the prevalence of chronic pulmonary aspergillosis in Vietnam. This is a good overall study from a country with little surveillance data. What the study is lacking is guidance from the authors. Based on their study, what are their recommendations for clinicians? Based on their study, how should clinicians in Vietnam be looking for CPA? Is there a diagnostic algorithm that can be followed based on the findings of these studies? Now that we have learned about CPA in Vietnam, how can we apply what we have learned?

Specific comments:

Abstract line 1: change to: chronic pulmonary aspergillosis

Line 152-153: What is the meaning of an internationally high rate? Do you high when compared to the rates in other countries or do you mean the highest rate amongst countries who have reported or calculated a rate?

Author Response

Manuscript ID: jof-1213898

Response to Reviewers

Dear Mr/Ms Reviewer,

 Thank you for giving us the opportunity to submit a revision of the manuscript

Type of manuscript: Article Title: “Chronic Pulmonary Aspergillosis situation among post tuberculosis patients in Vietnam: An observational study “ form publication in the Journal of Fungals.

We appreciate the time and the hard-working process that you and the reviewers dedicated to providing feedback on our manuscript and are grateful for the insightful comments on and valuable improvements to our paper. We have incorporated most of the suggestions made by the reviewers. Those changes are highlighted within the manuscript. Please see below, we have performed a point-by-point response to the reviewers’ comments and concerns. All page numbers refer to the revised manuscript file with tracked changes.

Reviewer 2 Report

The topic is quite interesting but the presentation is not convincing. What new message of this observational study is added to the actual knowledge of the diagnosis or pathogenesis of Aspergillus infection in patients with prior tuberculosis? Literature no3 has already provided quite similar information.

line 37; "the disease silently destroys lung" . Does this holds true for all entities of CPA? For example in patients with a fungus ball in the cavities there is no fungal tissue invasion .

line 43: how a fungal infection increases the risks of development of MDR mycobacteria??? (by the way in line 69 it has been stated that patients with active TB have been excluded from this study).

line 61: "prolonged respiratory symptoms" not precise; what the authors mean?

line 63: how bronchiectasis has been diagnosed? by radiological methods?

line 67: "or pathological evidence" not precise; what the authors mean?

line 71: only mycologic cultures (which one?) or even bacteriologic examination including Mycobacterium tuberculosis  as well as MOTT?

line 75: a description of the identification or differentiation mthods of Aspergillus fumigatus is lacking. An antimykogramme is not reported. What is the in vitro susceptibility to itraconazole?  (in general very low!)

line 75/76 "pathology" not precise; what the authors mean? positive bacterial culture? of histologic alterations?

line 78-81: the radiologic results should be prsented gefor the microbiologica data.

line 88: were the serum consentrations tested only once in a patient? why not examination of the dynamic of IgG ? why not galactomannan in BAL or serum was examined, which generally gives some evidence of presence of Aspergilli? Probably this assay would be mor relevnt than IgG determination. Why not the simple lateral flow assay (LAF) for A. fumigatus?

line 90: itraconazole orally?  tablets or liquid. The resorption largely depends on circumstantial factor such as food intake. Hence the serum levels achieved are rather variable!

Why only the minority of diagnosed patients where treated with itraconazole (which will be insufficient !). Which therapy received the majority? Nothing? Why a diagnosis of fungal infections should be made, when there will be no consequences????

line 111: why recurrency of tuberculosis occured?  not information about the treatment schedule is given. Are  MDR responsible?

line 114: again no information why  such a long treatment (over decades!) had been necessary; wrong drugs?

line 117: did hemoptysis also occured in patients with only fungal balls?? unprecise

line 119: dynamics of CRP under antifungal treatment?

line 122: "pleural thickening" because of funagl invasion? or rather due to residual alterations after chronic mycobacterial inflammation?

Table 1: blood test are listed under the heading of clinical symptoms???

No information about the  ranges of WBC and CRP. The numbers given mean those patinets with elevated levels??

Confusion: IgG (global IgG or anti-Aspergillus IgG??) Difficult to believe that only 1.5% of patients have IgG in their serum!!!!

Fig. 1 and 2 present the same data only in different depiction. No information about a correlation of anti -aspergillus IgG and the different clinical entities of CPA; do patients with fungus balls only develop serum IgG????

line 150 and 157: identical words

line 168: banal; can be omitted

line 174: obove it has been stated that patients with active tuberculosis have been exlcuded from this study.

line 180: this recommendation cannot be generalized

line 180-187 redundant

line 192-194: speculation

line 196-201 akward

line 203: is hemotysis typical for patients with fungus ball??

line 215: is there any evidence (uistology?) that aspergilli are rsponsible the "pleural parenchymal features" (what does the aithors mean by this term??)

line 224: indeed serology is good for "tracking the disease`s progression": but why the authors have not examined the dynamics of IgG in their patients?

line 228: yes citation no. 3 has already reported similar data; what is new?

line 229-233: this paper does not add any information on this aspect ; this comments could be omitted

line 234-239: poor data

line 240-243: irrelevant for this mansucript

line 244: many limitations

line 218: indeed these findings are already well known and have been reported before (no. 36,37 even in 1995!).

Author Response

Manuscript ID: jof-1213898

Response to Reviewers

Dear Mr/Ms Reviewer,

 Thank you for giving us the opportunity to submit a revision of the manuscript

Type of manuscript: Article Title: “Chronic Pulmonary Aspergillosis situation among post tuberculosis patients in Vietnam: An observational study “ form publication in the Journal of Fungals.

We appreciate the time and the hard-working process that you and the reviewers dedicated to providing feedback on our manuscript and are grateful for the insightful comments on and valuable improvements to our paper. We have incorporated most of the suggestions made by the reviewers. Those changes are highlighted within the manuscript. Please see below, we have performed a point-by-point response to the reviewers’ comments and concerns. All page numbers refer to the revised manuscript file with tracked changes.
Please see the attachment

Reviewer 3 Report

The authors have assembled most likely the first detailed report of the incidence of chronic aspergillosis in a well-characterized cohort in Vietnam.  An important finding from this study is the extraordinarily high incidence of CPA in Vietnam, indicative of, if not a crisis, a severely under-recognized condition in need of improved diagnostic strategies and aggressive management.  This is an important message to transmit to the community involved in fungal disease research and clinical management.

Major comments:

  1. Thorough English editing is needed
  2. Figure 2 data, while statistically significant are clinically meaningless as the correlation line is nearly horizontal and the data overall appear to be a random assemblage of points on the matrix. An R2 correlation coefficient value is also not included…and would be expected to be extremely low.  Rather than include these data, which do not strongly support the report even if they were more robust, suggest replacing them with select plain or high resolution CT imaging of selected patients illustrating the types of Aspergillosis-related disease as seen in the cohort.

Especially since the findings indicate a much wider disease prevalence, when combined with the findings of others showing a high incidence of antifungal resistance among aspergillus isolates, the findings overall suggest that Vietnam needs to launch an aggressive public health campaign centered on CPA, if this has not already happened.  If perhaps such a campaign is at least in the planning stages, inclusion, at least in outline form, of such a campaign in the Discussion section would strengthen the manuscript.

Author Response

(The authors gave the same response as above.)

Round 2

Reviewer 2 Report

There are still some inconsistencies in the manuscript

List of authors: The city of the National Lung Hospital is lacking

line 92: resorption of intraconazole  is dependant on a low gastric pH; hence, it is generally recommended to take the drug just after a meal. You have not mentioned if your patients have respected this fact.

line 256: this sentence should be improved

line 263: ref. 37 is rather old

line 276: using ? better based on

Table 1: history of smoking? no information; Chest CT findings should be place in the middle of the page like other headings

Fig. 1:  A description of the vertical axis (ordinate) is lacking

Fig 3: you stress upon the fact that arteries are enlarged. Furthermore, you find often pleural thickening. But in the discussion there is no explanation for the pathogenesis of these observations. Sequelae of prior TB???

Author Response

Dear Mr. Reviewer,
Thank you very much for your kind support. I have learned a lot of valuable knowledge from your dedicated work. I would like to submit my revision.  Please see the attachment.
Have a nice weekend.

Reviewer 3 Report

Manuscript is now much improved.

Author Response

Thank you for your dedication and kind support.